# The Use of Intraoperative Microvascular Doppler in Vascular Neurosurgery: Rationale and Results—A Systematic Review

**DOI:** 10.3390/brainsci14010056

**Published:** 2024-01-06

**Authors:** Vincenzo Gulino, Lara Brunasso, Chiara Avallone, Benedetta Maria Campisi, Lapo Bonosi, Roberta Costanzo, Emanuele Cammarata, Carmelo Lucio Sturiale, Adriana Cordova, Domenico Gerardo Iacopino, Rosario Maugeri

**Affiliations:** 1Neurosurgical Clinic AOUP “Paolo Giaccone”, Post Graduate Residency Program in Neurologic Surgery, Department of Biomedicine Neurosciences and Advanced Diagnostics, School of Medicine, University of Palermo, Via del Vespro 127, 90127 Palermo, Italy; vincenzo.gulino@community.unipa.it (V.G.); lara.brunasso@community.unipa.it (L.B.); chiara.avallone@community.unipa.it (C.A.); benedettamaria.campisi@community.unipa.it (B.M.C.); lapo.bonosi@community.unipa.it (L.B.); roberta.costanzo@community.unipa.it (R.C.); rosariomaugeri1977@gmail.com (R.M.); 2Plastic and Reconstructive Surgery, Department of Surgical, Oncological and Oral Sciences, University of Palermo, Via del Vespro 127, 90127 Palermo, Italy; manu.cammarata@yahoo.it (E.C.); adriana.cordova@unipa.it (A.C.); 3Department of Neurosurgery, Fondazione Policlinico Agostino Gemelli IRCCS, Largo Agostino Gemelli 8, 00168 Rome, Italy; carmelo.sturiale@policlinicogemelli.it

**Keywords:** microvascular Doppler sonography, flowmetry, vascular neurosurgery, intracranial aneurysm, arteriovenous malformation, arteriovenous fistula, intraoperative setting, clipping

## Abstract

Surgical treatment of neurovascular lesions like intracranial aneurysms, arteriovenous malformations and arteriovenous dural fistulas is still associated with high morbidity. Several recent studies are providing increasing insights into reliable tools to improve surgery and reduce complications. Inadvertent vessel compromise and incomplete occlusion of the lesion represent the most possible complications in neurovascular surgery. It is clear that direct visual examination alone does not allow to identify all instances of vessel compromise. Various modalities, including angiography, microvascular Doppler and neurophysiological studies, have been utilized for hemodynamics of flow vessels in proper clipping of the aneurysm or complete obliteration of the lesion. We intended to review the current knowledge about the intraoperative microvascular Doppler (iMDS) employment in the most updated literature, and explore the most recent implications not only in intracranial aneurysms but also in neurovascular lesions like arteriovenous malformations (AVMs) and arteriovenous dural fistulas (AVDFs). According to the PRISMA guidelines, systematic research in the most updated platform was performed in order to provide a complete overview about iMDS employment in neurovascular surgery. Twelve articles were included in the present paper and analyzed according to specific research areas. iMDS employment could represent a crucial tool to improve surgery in neurovascular lesions. The safety and effectiveness of the surgical treatment of neurovascular lesions like intracranial aneurysm and other neurovascular lesions like AVMs and AVDFs requires careful and accurate consideration regarding the assessment of anatomy and blood flow. Prognosis may depend on suboptimal or incomplete exclusion of the lesion.

## 1. Introduction

The safety and effectiveness of the surgical treatment of neurovascular lesions like intracranial aneurysms and other neurovascular lesions like arteriovenous malformations (AVMs) and arteriovenous dural fistulas (AVDFs) require careful and accurate consideration regarding the assessment of anatomy and blood flow. Prognosis may depend on suboptimal or incomplete exclusion of the lesion. Despite all recent advances in microvascular surgery, nowadays intracranial aneurysm represents a clinical entity still associated with high morbidity [1]. Successful results hang between the complete aneurysm occlusion and the preservation and integrity of parent arteries. Indeed, the outcome may be adversely impacted by local cerebral ischemia from inadvertent occlusion of adjacent vessels following aneurysm clip ligation. Therefore, it is crucial to verify aneurysm occlusion and to avoid inadvertent stenosis of neighboring vessels during surgery. It remains evident that in most cases intraoperative visual inspection is inadequate in verifying the correct placement of aneurysm clips.

The correct placement of the clip may be verified intraoperatively by several methods such as digital subtracting angiography (DSA), microvascular Doppler sonography (iMDS), near-infrared indocyanine green angiography (ICG-VA), intraoperative neuromonitoring with somatosensory evoked potentials (IOMN) and endoscopy. Among these methods, the intraoperative microvascular Doppler (iMDS) combined with miniaturized probes seems to represent the most simple and reliable method to verify blood flow velocity. It allows the neurosurgeon to assess atraumatically and immediately the blood flow velocity, verifying proper clip placement and the maintenance of adequate blood flow in the adjacent vessels. In this paper, our aim is to evaluate the usefulness and reliability of intraoperative microvascular Doppler sonography in neurovascular surgery, first verifying employment in intracranial aneurysms. Moreover, the authors give a panoramic view of the employment of intraoperative microvascular Doppler in AVM and AVDF.

## 2. Materials and Methods

A systematic literature review was conducted according the Preferred Reporting Items for Systematic Reviews and Meta-Analyses (PRISMA) (not registered) guidelines on PubMed, Web of Science, the Cochrane Library and Scopus databases [2]. This review was not recorded on prospective registers, thus review protocol was not prospectively available. No specific timeframe was considered for the literature review, and the following search string were searched from database inception to October 2023: ((“microvascular doppler”) OR (“micro probe”) OR (“doppler”) OR (“intraoperative probe”) OR (“intraoperative doppler”) OR (“microflow probe”)) AND ((“vascular neurosurgery”) OR (“microsurgical clipping”). An author (V.G.) screened abstracts for eligibility and conducted the first phase for including relevant papers about the topic. Collected data were evaluated by three authors (V.G., C.A. and B.M.C.). Disagreement over the issue was solved with discussion with a supervisor (L.B.). 

Duplicated papers were removed using Microsoft Excel 16.37 Software (Redmond, WA, USA). Titles and abstracts were screened and nonrelated articles were excluded. Original articles, prospective and retrospective cohort studies, case series and technical notes reporting the use of microvascular Doppler in the neurovascular surgery field were considered eligible for the literature review. Case reports, review articles and nonEnglish papers were excluded and removed. Each articles’ full text was retrieved for further examination when title and abstract corresponded to the screening requirements. The following data were extracted from the included studies: number of patients, type of vascular lesion, study design, iMDS features, other intraoperative tools than iMDS, vessels characteristics, type of iMDS and outcomes. 

## 3. Results

The literature search provided 301 results and 66 duplicates were removed before topic screening. From 235 inherent records screened, 208 papers were excluded by title, 13 by abstract and 2 because of nonEnglish language. Finally, 12 papers were eligible for this review and included and analyzed for the present work (Figure 1). Full text was available for all. All papers are in the details presented in Table 1. 

A total of 12 retrospective clinical studies including vascular lesions treated were retrieved. Eight papers employed iMDS in surgical treatment of intracranial aneurysms. Two papers evaluated lumbar or thoracic dAVFs and another two studies assessed iMDS in brain AVMs. A cumulative number of 711 patients (male-to-female ratio 1:1.71, mean age: 56 ± 7.7 years) with 764 vascular alterations were included in this study. A total of 686 out of 764 (89.8%) were anterior and posterior circulation aneurysms, 59 out of 764 (7.7%) were AVMs and 19 out of 764 (2.5%) were spinal dAVFs. In 6 [3,4,5,6,7] out of 12 cases intraoperative microvascular doppler sonography was obtained with a 1 mm diameter microprobe employing an ultrasound frequency of 20 MHz. Two papers [4,8] set insonation angle at 30–60°. One study [9] equipped a transcranial doppler ultrasonography machines with a 20 MHz ultrasound microprobe. Three papers [10,11,12] showed a blood flow velocity (BFV) assessment with the Charbel Micro Flowprobe. A total of 6 of 12 papers did not examine or compare anything other than iMDS techniques while ICG-VA was always employed in rest of studies. Other technical aspects of iMDS methodology, the anatomic location of the aneurysms, different employed methods for verifying clip positioning or assessing blood flow velocity, the accuracy and outcomes of the intraoperative iMDS are summarized in Table 1. 

Few studies verified accurate positioning of the clip through iMDS. Clip position was inappropriate in 30 of the 123 aneurysms (24.4%) treated by Gruber [5], 10.6% in the paper by Cui [8], 23.8% in Hui et al. [4], 22% in the Pereira et al. [6] study. iMDS employment showed complete aneurysm exclusion after modified position of the clip in all cases assessed by Cui, in eight (7.9%) cases evaluated by Hui. In giant aneurysms the clip was repositioned based on iMDS findings in 11 out of 14 (78.6%) aneurysms [4]. No repositioning of the clip was necessary as a result of BFV measurements performed in 39 patients described by Malinova et al. [3]. Doppler recorded a decreased blood flow >25% of the basal flow measurement in 24 of 95 (25.2%) patients evaluated by Della Puppa et al., 2018 [10]. 

iMDS documented absent parent artery flow after clipping of the aneurysm in 4 out of 15 cases assessed by Gruber [5] and in all 11 cases assessed by Della Puppa et al., 2014 [10]. A stenosis of vessels induced by the clip was detected in 10 out of 79 (12.7%) cases in paper by Cui [8]. Moreover, iMDS identified increased blood flow velocity in the parent artery in 16 of 101 cases (15.8%) assessed by Hui et al. [4]. 

A study conducted by Pereira et al. [6] showed that ischemic infarcts rate in group of 50 patients was 4% (2 out of 50 cases), compared to 9 of the 50 cases (18%) evaluated in group without use of iMDS (*p* = 0.02).

Assessing the effect of combining application of intraoperative IOMN, microvascular Doppler sonography and ICG-VA, Li et al., 2016 [13] demonstrated that iMDS detected 9 cases (5.7%) of somatosensory evoked potential (SSEP) amplitude changes and a greater than 10% increase in flow velocity of parent arteries. In all cases, SSEP and iMDS returned to baseline levels after the aneurysm clip was repositioned. 

Burkhardt et al., 2015 [7] and Della Puppa et al., 2016 [12] assessed BFV in brain AVMs in 32 and 27 cases, respectively. The measurements recorded by Burkhardt revealed systolic flow velocities of 7 to 22 cm/s (mean 16 cm/s) and diastolic values of 5 to 12 cm/s (mean 10 cm/s) in the feeding arteries. Drainage vein velocities were measured at 16 cm/s (systolic) and 12 cm/s (diastolic). Turbulences and pulsating superimpositions were seen in both feeders and arterialized drainage veins in all 32 patients. The mean PI (<0.72) and mean RI (<0.5) were significantly lower compared with the standard values for normal cerebral arteries. No flow signals could be obtained from the feeding vessels after surgical resection of the malformation in the first study, while a residual flow of the venous drainage was detected in 5 out of 27 cases (18.5%) described by Della Puppa [12]

Two studies described iMDS employment in spinal dAVFs. Della Puppa and colleagues, 2016 [14] showed that blood flow in perimedullary venous plexus ranged from 2 to 12 mL/min (mean 8 mL/min). BFV always dropped lower than 1 mL/minute after temporary and definitive fistula occlusion by clip positioning and coagulating respectively, in all cases. Flowmetry assessed by Xing et al., 2019 [9] in seven patients with treated spinal DAVFs showed an arterial feeder flow ranging from 3 to 20 mL/min, (mean 11 mL/min). Even in this case, a flow lower than 1 mL/min was recorded by micro Doppler after temporary and definitive clip positioning.

**Table 1 brainsci-14-00056-t001:** Qualitative analysis of the main characteristic of the articles included in the present systematic review. Abbreviations: iMDS = intraoperative micro Doppler sonography; R = retrospective; ICG-VA = indocyanine green video angiography; DSA = digital subtraction angiography; IONM = intraoperative neuromonitoring; SSEP = somatosensory evoked potentials; AcoA = anterior communicating artery; ACA = anterior cerebral artery; ICA = interior cerebral artery; PCoA = posterior communicating artery; AVM = arteriovenous malformation; BF = blood flow; BFV = blood flow velocity; OA = ophthalmic artery.

Author, Year of Publication	N Patient/ Lesions	Study Design	iMDS Characteristics	Other Intraoperative Methods	Vessel Characteristics	iMDS Employment	Outcomes
Xing XZ et al., 2019 [9]	7	R	Transcranial Doppler ultra-sonography machines (Viasys Healthcare, USA). Probe diameter: 1 mm, Frequency: 20 MHz	None	2 lumbar and 5 thoracic spinal arteriovenous fistulas.	Flow measurement before and after dural opening, during temporary clip occlusion when considered relevant, after final obliteration, on venous drainage	Final flow measurements showed arterial spectrum flow disappeared in all cases.
Della Puppa A et al., 2018 [10]	85/96	R	(Charbel Microflowprobe; Transonic System, Ithaca, NY, USA)	IONM, ICGA-VA	46 ruptured, 50 unruptured aneurysms. Proximal group (ICA, A1, M1) (45); Distal group (pericallosal, MCA bifurcation), (51)	Flow measurement before and after clipping to exclude a flow drop and verify flow preservation.	Clip repositioning performed 24 times according to blood flow impairment (24/42; 57.1%). Decreased flow recorded in 24 patients.
Della Puppa A et al., 2016 [12]	12	R	(Charbel Microflow probe; Transonic Systems, Inc., Ithaca, NY, USA).	ICGA-VA	10 thoracic and 2 lumbar spinal arteriovenous fistulas.	Identification of fistula point, assessment of final fistula point exclusion, flow measurement before and after temporary/permanent occlusion of the fistula.	Flow drop in all procedures, confirming complete fistula occlusion. Flowmetry presents a sensitivity similar to ICG-VA and helps to clarify ICG-VA.
Della Puppa A et al., 2016 [11]	27	R	Charbel Micro Flowprobe, Transonic Systems, Inc., Ithaca, NY, USA).	ICGA-VA, temporary artery clipping test	21 ruptured, 6 unruptured brain AVMs	Flow assessment before resection, after ICG-VA, during dissection on the same vessels or on different vessels and after surgical resection	iMDS was helpful in making ICG-VA data clearer before resection in 22% of cases. No AVM remnants reported at postoperative DSA when flowmetry was performed. iMDS was more reliable than ICG-VA in detecting residual nidus missed at resection of deep or subcortical/interhemispheric AVMs.
Li Z et al., 2016 [13]	158	R	DIGI-LITE Doppler blood flow analyzer. Probe diameter: 1 mm; Frequency: 16 MHz; M-mode depth bounds of 0.8~13.6 mm for doppler monitoring	SSEP, ICGA-VA	ICA (36.7%); ACoA (31%); MCA (16.5%); OA (13.3%); ICA bifourcation (2.5%) aneurysms	Monitoring BF spectrum and velocity of proximal and distal fragments of parent artery	Perforating vessel occlusion in 10 patients with MCA a. >10% increase in flow velocity of parent arteries in 12 patients with ACoA a. and 8 patients with OA a.
Pereira BJ et al., 2015 [6]	50	R	Mizuho Surgical Doppler (Mizuho Corporation, Tokyo, Japan). Frequency: 20 MHz	None	MCA (21), ACA (9), PCoA (9), ICA (9)	Cerebral flow assessment. Data compared with a control group without iMDS.	Information provided by iMDS influenced clip repositioning in 11 patients (22% of cases). Ischemic infarct rate (4% in the group with iMDS vs 18% in the group without iMDS).
Burkhardt T et al., 2015 [7]	32	R	DWL SmartDop®, Compumedics, Singen, Germany. Probe diameter: 1 mm, Frequency: 16 MHz	None	brain AVM	Systolic- and diastolic-flow velocity main feeders and drainage vessels of AVM priorly verified by preoperative DSA	Normalized venous flow patterns without arterial flow turbulences at the end of the surgical procedure in all 32 cases
Malinova V et al., 2015 [3]	39	R	DWL SmartDop®, Compumedics, Singen, Germany. Probe diameter: 1 mm, Frequency: 20 MHz. Probe coupled with a neuronavigational pointer	None	29 unruptured, 10 ruptured a. MCA (23); ACoA (7); ICA (6); PCoA (2); ACA (1) aneurysms	BVF measurement before and after aneurysm clipping	Mean deviation of the BFV before and after clipping was only 2.12 cm/s. NO BFV drop, no need of repositioning of clipping. A postoperative hemiparesis occurred due to occlusion of a perforating vessel not monitored by iMDS.
Della Puppa A et al., 2014 [11]	26/34	R	Charbel Micro Flowprobe, Transonic Systems, Inc., Ithaca, NY, USA).	ICG-VA	MCA (21), ACA (8), ICA (5); 25 unruptured, 9 ruptured.	BFV pre and postoperative assessment. A drop >25% in postclipping blood flow indicates a clipping repositioned.	No flow reduction recorded after clip reposition. Flow reduction >25% in 8 out of 48 cases (16%).
Hui PJ et al., 2013 [4]	92/101	R	Doppler Blood Flow Monitoring System (Companion III, Germany). Probe diameter: 1.5mm, Frequency: 20 MHz. Insonation angle: 30–60°	None	82 aneurysms ligated with a single clip; 19 paraclinoid, ophthalmic segment, large or giant aneurysms with two or three clips.	BFV before and after the clip application. Data compared with a control group without iMDS	iMDS identified compromised blood flow due to inaccurate clip placement in 11 of the 101 (10.9%) aneurysms. Stenosis of the parent artery in 19 cases (18.8%), increased of BFV in parent artery in 16 of these cases (15.8%). No residual aneurysm or vessels stenosis in iMDS group.
Cui H et al., 2011 [8]	79/85	R	Companion III (SciMed, Bristol, UK). Probe diameter: 1mm, Frequency: 20 MHz. Insonation angle: 30–60°	None	50 aneurysms <10 mm; 24 between 10 and 20 mm, 11 aneurysms >20 mm	BFV before and after the clip application	Incomplete occlusion in 9/85 a. (10.6%). Vessel stenosis induced by the clip in 10 of 79 cases (12.7%).
Gruber A et al., 2011 [5]	104/123	R	DWL SmartDop®, Compumedics, Singen, Germany. Probe diameter: 1mm, Frequency: 20 MHz	ICGA-VA, DSA, endoscopy	AcoA (31), ACA (11), ICA (15), MCA (63), posterior circulation (3) aneurysms	Assessment of the parent arteries before and clipping	Absent parent artery flow documented in 4 out of 15 cases.

## 4. Discussion

The Doppler effect occurs when the frequency of a transmitted ultrasound changes due to its reflection in a moving object. Intraoperative micro-Doppler sonography is a technique based on high frame rate ultrasound and subsequent Doppler processing that enables real-time detection of microvascular hemodynamics with submillimeter and subsecond precision [15]. When the technique is applied to vessels, the difference in frequency of transmitted and received ultrasound is proportional to the blood flow velocity [16]. 

Doppler sonography was first introduced for the assessment of cerebral hemodynamics in extracranial arteries in 1965 [17]. It was useful for instantaneous assessment of dynamic changes of the blood flow, resulting from various stresses to circulatory system and for an objectively and simply evaluation of cerebral circulation and arteriosclerosis. Twenty years later, Aaslid et al. [18] modified this technique for a noninvasive transcranial method for determining flow velocities in the basal cerebral arteries. Nornes [19] in 1979 and Gilsbach [20] in 1983 first applied the transcranial doppler ultrasonography methods to the intraoperative evaluation of small cerebral arteries. Nornes used a pulsed echo Doppler flowmeter operating at 6 mHz, with gas-sterilized miniature ultrasound probes. Gilsbach used a 20 MHz microvascular Doppler system for intraoperative use and a transcranial Doppler at 2 MHz for the postoperative evaluation. In this scenario, technical progress led to the progressive development of small Doppler probe of 0.3–1 mm in diameter and with a high frequency of 20 MHz that allowed a direct intraoperative measurement of Doppler signal of small vessels, such as perforating arteries. 

The microflow probe consists of an electronic flow detection unit with a flow sensing probe that is used outside the vessels. The flow probe uses the principle of ultrasonic transit time to sense flow in vessels [21]. The flow in milliliters per minute appears on the detection unit as a digital display and indicates positive or negative waves dependent on the direction of flow [22]. Different types of probes are now available, and the main differences consist in the shape of the instrument and the extent of vascular dissection. For the flow measurement, a portion of the vessel of interest is dissected free from the surrounding arachnoid. For the Charbel probe, the probe is hooked around the vessel under direct vision, so there is the need for a circumferential dissection of the vessel of interest [21]; the main limitation is that measurements require further and unnecessary dissection in fragile venous vessels that can be potentially injured during the maneuver, adding an extra risk to the surgical procedure. To overcome these difficulties and limitations, new linear-shape probes have been developed that can be just gently applied on the vessel wall at a specific angle without compression to detect blood flow and direction. All this has been added to the possibility of measuring flow velocity, in order to carry out not just a qualitative measure of the blood flow but a more through evaluation. 

Nowadays, microvascular Doppler sonography represents a widely used method in neurovascular surgery. Several authors has described their technical experiences and applications for the assessment of blood flow velocity in order to ensure the best postoperative patients’ outcome. Especially in aneurysm surgery, the intraoperative application of microvascular doppler sonography has become paramount for verifying proper clip placement and the maintenance of adequate blood flow in collateral arterial vessels [23,24,25,26,27,28]. While the application of intraoperative DSA presents a number of difficulties in diffusion, microvascular Doppler sonography is a minimally invasive, real-time, sensitive, and easily reproducible technique complementary to intraoperative ICG-VA [29] and neurophysiologic monitoring.

### 4.1. Use of Intraoperative Microdoppler Sonography in Aneurysm Surgery 

The main challenge of aneurysm surgery is to exclude the aneurysm from the intracranial circulation while maintaining the patency of normal surrounding vessels and ensuring the equivalent blood flow. Inaccurate clip application could be associated with compromised blood flow and subsequent development of delayed neurological deficit, stroke, and occasionally death. Minimizing ischemic injury during microsurgical clipping remains a goal in the learning curve of a neurovascular surgeon. Although advanced preoperative and intraoperative imaging techniques are available [30], real-time intraoperative assessment of the blood flow and hemodynamics remains a crucial point. iMDS provides a functional and noninvasive intraoperative examination that helps in identifying the aneurysmal peri-saccular vessels pattern, even under complicated anatomic conditions. This is particularly important in the context of an increasing mini-invasive surgery, after the introduction of the minipterional variant approach for unruptured middle cerebral artery aneurysms, and the miniaturization of the Sylvian fissure opening through a tailored opening [31,32,33,34,35]. A representative illustration is shown in Figure 2.

In the most updated recent literature, several studies have demonstrated the usefulness of microvascular Doppler sonography in the routinely aneurysm surgery [20,36,37]. Before the aneurysm is clipped, micro Doppler can help in identifying the aneurysm pattern by detecting the direction and hemodynamics of parent arteries and collateral vessels [36]. As described by Li [13] and Hui [4] in their papers, the detection of vortex flow or thrombus in the aneurysm sac through the waveform and the acoustic signal represents a useful tool, especially when aneurysm is not clipped yet. After the positioning of the clip, micro Doppler can help in assessing whether the inflow to the aneurysmal sac is completely excluded, and whether the patency of parent and collateral arteries which can be stenosed or closed by mistake. Incomplete aneurysm exclusion is documented by persistent blood flow through the aneurysmal dome after clip placement; it was detected by iMDS in 9 of the 85 cases reported (10.6%) by Cui et al. [8], according to data reported by Hui [4] (10.9%). Information provided by iMDS influenced clip repositioning in 11 patients (22% of cases) in study of Pereira [6]. No residual blood flow in verifying correct aneurysm exclusion was showed by Della Puppa [10] after the microprobe employment. In addition, the technological update of this tool also allows to evaluate the blood flow of the examined vessel together with flow direction, and this is paramount in identifying an early recognition of a potential misdiagnosed vessel stenosis after the clip positioning, or vasospasm with a high degree of sensitivity [23]. 

Another crucial aspect in aneurysms surgery is represented by the time required for verifying proper clipping of the aneurysm and confirming parent artery patency. It requires no more than few minutes, as described in most of studies [4,7,25,38,39]. Considering other available alternative modalities that requires 30 to 60 min to be performed, this interval may be too long to avoid cerebral ischemia or infarction. Moreover, in complex aneurysms requiring repeated clip adjustments, quickly repeatable assessment of arterial patency is a necessary procedure. 

In this review, few studies compared the accuracy and the sensitivity of iMDS to those of digital subtraction angiography (DSA) or ICG-VA. As demonstrated by Bailes et al. [25] and Woydt et al. [40], iMDS could represent a reliable and efficient tool for the assessment of cerebral hemodynamics in aneurysm surgery. Even Kapsalaki et al. [41] reported 98% accuracy of micro Doppler compared to postoperative DSA in their study. Another point of view is given by Gruber and colleagues [5]. They reported that the intraoperative monitoring and vascular imaging methods are complementary rather than competitive, and none of the devices used is absolutely reliable as a stand-alone method. In fact, iMDS documented absent flow in the parent artery in 4 out of 15 aneurysms treated. The remaining parent artery stenoses not detected by iMDS were found in seven cases by ICG-VA. Moreover, Li et al. [13] stress a combining application of real-time ICG-VA and iMDS to significantly reduce the residual rate of aneurysm neck and improve the quality of aneurysm clipping. Somatosensory evoked potentials are another technique that can be used intraoperatively during aneurysm surgery [42,43] but they are strongly influenced by the anesthetic procedures and can detect the occurrence of only major cerebral ischemia. Definitely, the only way to intraoperatively assess complete aneurysm occlusion and parent artery patency is the combination of these techniques.

Another aspect to be mentioned is the availability of the tool. Despite all other techniques such as iDSA, ICG-VA and neurophysiological monitoring, most of the neurosurgical operating rooms are equipped with microvascular probes, and the intraoperative application of iMDS does not requires special training. Moreover, the total cost of the device is quite low compared to the equipment required, in example, for performing DSA. 

Several technical limitations need to be considered for the iMDS. First, iMDS is an operator-dependent technique, and this can significantly influence the measurements and findings, especially in the case of small-caliber vessels. Few authors described aneurysm remnant or a residual neck that may go unnoticed with iMDS assessment [23,25,39,44], especially in case of wide neck and fusiform aneurysms [4]. As reported by Malinova et al. [3], the measurement of blood flow vessels depends both on the insonated vessel segment as well as on the insonation angle; it can be difficult to maintain the specific insonation angle at the insonated segment before and after clip placement. To solve this problem, authors [3] employed the use of neuronavigational screen for reproducibility. Unfortunately, brain shift caused by intraoperative loss of cerebrospinal fluid can have a negative influence on the accuracy of the neuronavigation. Finally, measurements require further dissection in fragile vessels that can be potentially injured during the maneuver [10].

### 4.2. Use of Intraoperative Microdoppler Sonography in Arterio-Venous Malformations Surgery 

Arteriovenous malformations (AVMs) are congenital vascular lesions characterized by high-velocity arterial and venous shunting [7]. The annual bleeding rate of intracranial AVMs is estimated to be between 2% and 4% [45], depending on the size and location of the AVMs, the type of venous drainage and any associated aneurysms. Imaging the vascular morphology of the AVMs is a critical step in surgical planning, intraoperative decision making, and ultimately treatment success [46,47]. Over the years, several techniques have been developed in this field for intraoperative monitoring during this type of procedure and, as in intracranial aneurysm surgery, the most common are iDSA and ICG-VA [7]. Although intraoperative flow assessment with micro Doppler sonography is a validated technique in aneurysm surgery to detect flow reduction during microsurgical clipping, the literature and the authors experience regarding the use of this tool in both cerebral and spinal AVMs is limited. 

In the present systematic review, two articles [7,14] reported the use of iMDS during cerebral AVMs surgery. Burkhardt et al. [7] measured intraoperative flow using micro Doppler in a case series of 32 consecutive patients. After surgical exposure of the AVMs, ultrasound examination of both the afferent and efferent vessels was performed before and after surgical resection of the malformation. In the authors’ experience, micro Doppler revealed systolic flow velocities of 7 to 22 cm/s (mean: 16 cm/s) and diastolic values of 5 to 12 cm/s (mean: 10 cm/s) in the feeding arteries. In all 32 patients, turbulence and pulsatile superimposition were observed in both feeder and arterialized drainage veins. 

Della Puppa and colleagues [14] reported a multimodal flow-assisted approach for the treatment of 27 cerebral AVMs based on the concomitant use of ICG-VA, microflow probe flowmetry. Temporary arterial clipping test under intraoperative monitoring was also described. In this report, the Charbel microflow probe was used before resection on both cortical feeders and venous drainers, during dissection on the same vessels and finally at the end of surgical resection on the venous drainage before section to evaluate residual flow. Although the deep location of the venous drainage restricts flow assessment using the microflow probe, in all cases iMDS was helpful in understanding the angioarchitecture of the AVM and in guiding surgical planning. Indeed, they defined the multimodal flow-assisted approach to AVM surgery as a feasible, safe and reliable method to achieve AVM resection with high radicality and low morbidity.

### 4.3. Use of Intraoperative Microdoppler Sonography in Dural Arterio-Venous Fistula Surgery

Type I spinal arteriovenous malformation, also known as spinal dural arteriovenous fistula (dAVF), is an arteriovenous shunt between a radiculo-meningeal artery and a radicular vein with retrograde drainage to the perimedullary vessels, leading to venous congestion of the spinal cord [48,49,50]. Successful identification and complete obliteration of the fistula without compromising spinal venous drainage is the essence of spinal dAVF treatment [51]. Treatment options for spinal dAVF include endovascular treatment with fistula occlusion or microsurgical fistula occlusion, which has been shown to be the most definitive treatment with a low rate of recurrence. Accurate identification of the fistula and exclusion of the resident after obliteration remain a major challenge for patients with spinal dAVF, especially in line with the most recent mini-invasive surgical approaches and consequent small surgical fields [52,53]. To overcome this challenge, various techniques have been developed over the years, such as the use of ICG-VA, which has proven to be a useful tool in the surgical management of spinal dAVFs by helping to localize the fistula and confirm its closure [54]. iMDS has made its entry as an atraumatic and real-time blood flow assessment in guiding the surgical procedure of fistula exclusion [55]. 

The use of intraoperative microvascular Doppler for spinal dAVFs has been previously described [56,57,58,59]. iMDS can clearly identify fistula by its abnormal arterialized venous flow, elevated blood flow velocities and increase of the resistance indices [60,61]. In most cases, intraoperative visual inspection of the draining vein can be sufficient to confirm fistula obliteration. Sometimes, the angioarchitecture is complex and visual inspection together with the video angiography assessment may not be enough in correctly identifying the fistulous site. So iMDS can first guide the surgeon in following the flow until the fistulous point is identified. After that, a confirmation by ICG-VA can improve safeness, before coagulation and occlusion of the fistula. In this scenario, Della Puppa and colleagues [12] proposed a combination of using both iMDS and ICG-VA for quantitative flow measurements, especially in more challenging malformations. According to Xing and colleagues [9], micro-vascular Doppler can play a role in some stages of spinal dAVF surgery, when doubt may arise as to the surgical obliteration strategy, which is characterized by feasibility and lower cost and can be repeated anytime when needed. The use of iMDS supports the reduction of surgical incision and opening, minimizing the extent and invasiveness of the surgical approach and supporting better postoperative patient outcomes. 

## 5. Conclusions

Intraoperative microvascular doppler sonography represents a safe, feasible, noninvasive, instantaneous, effective, reliable and cost-effective method for documenting the patency of parent vessels, arterial branches and major perforators and the complete occlusion of cerebral aneurysms. Despite some limitations that makes iMDS complementary to other techniques, and considering the inevitable need for a multimodal approach to the control of vessels patency, this technique can be reliably used, in many instances, for the surgical treatment of aneurysms. Intraoperative MDS should be used routinely in neurovascular surgery, especially in cerebral aneurysm surgery and particularly for large, complicated and giant aneurysms. Indeed, it is reliable and safety use for other neurovascular pathology, like brain and spinal arteriovenous malformations.

## Figures and Tables

**Figure 1 brainsci-14-00056-f001:**
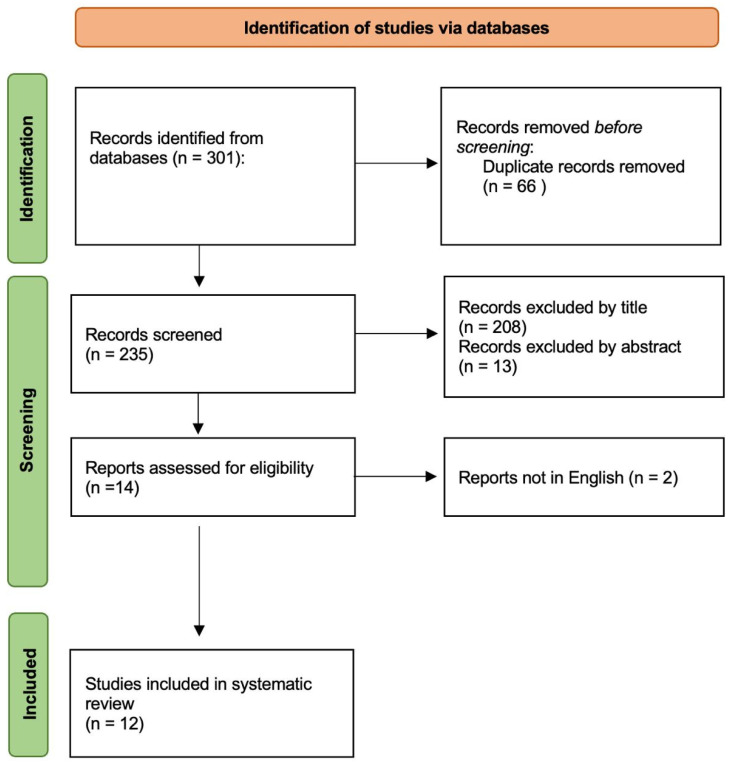
Flow diagram of the results of this systematic review according to PRISMA guidelines.

**Figure 2 brainsci-14-00056-f002:**
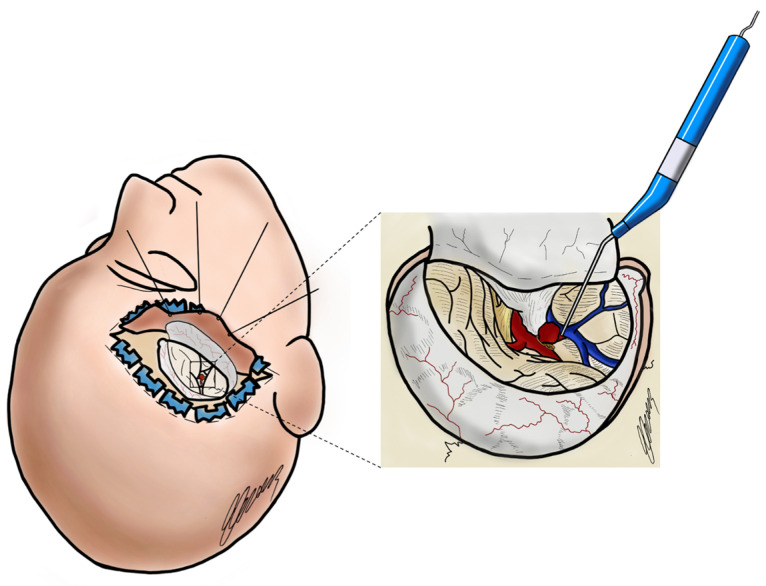
Graphic illustration by the author E.C. shows a minipterional approach for a right MCA aneurysm. Linear micro Doppler probe is represented during the evaluation of blood flow in all perivascular vessels after clip positioning.

## Data Availability

Not applicable.

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
