# Peer review of "The Use of Intraoperative Microvascular Doppler in Vascular Neurosurgery: Rationale and Results—A Systematic Review"

_brainsci, 2024, doi:10.3390/brainsci14010056_

Round 1
Reviewer 1 Report
Comments and Suggestions for Authors
I have read with pleasure this review about use of Intraoperative microvascular doppler in vascular neurosurgery. The literature is scarce and although probably more frequently used than published, the introduction of IcG probably overlapped discussion about other tools, except, maybe, intraoperative angiography (which is the gold standard in neurovascular surgery but not affordable for many). This review fills a hole in the promotion of standardized tools to improve the surgical results in vascular surgery, offering the possibility to control and correct the occlusion of aneurysms and other vascular malformations.
What is of outmost importance is the recommendation of combining different tools, as IcG and doppler to enhance the results of this delicate surgery, which requires a great deal of training and education, I agree with the authors conclusions.
I would suggest to separate the topic of AVMs and fistulas, where systematic data are even less and probably the need for angiography is certainly greater, even though the use of doppler might be generalized also in these surgeries.
There are some minor changes that would be welcome. The concept of "risky" surgery is somehow avoidable. Aneurysm closure by open surgery or endovascular treatment, carries the possibility of isquemic and hemorrhagic complications, as we all know, but the spontaneous rupture of an aneurysm itself is the worst risk of all. That's why neurovascular surgery (open or endo) should be performed by trained neurovascular surgeons, and they should be aware and able to use all those surgical tools that help to perform a secure surgery with the goal of occluding the aneurysm while leaving the parent and efferent arteries safe.
I believe this revision will help vascular neurosurgeons to recognize the applications and limitations of intraoperative doppler and the different systems available, as the benefit obtained by combining this tool to others as IcG.
Comments on the Quality of English LanguageMinor ortographic mistakes in the text; should be revised by a native english reader.
Author Response
Manuscript ID: brainsci-2774818
Title: The Use of Intraoperative Microvascular Doppler in Vascular Neurosurgery: Rationale and Results. A Systematic Review.
Authors: Vincenzo Gulino, Lara Brunasso, Chiara Avallone, Benedetta Maria Campisi, Lapo Bonosi, Roberta Costanzo, Emanuele Cammarata, Carmelo Lucio Sturiale, Adriana Cordova, Domenico Gerardo Iacopino, and Rosario Maugeri
Dear Editor and Reviewer 1,
First of all, I would like to thank you for your time and your suggestion.
I agree with you regarding the promotion of standardized tools to help vascular neurosurgeons recognizing the applications and limitations of intraoperative doppler. This is the main purpose of our revision.
Please find below a point-by-point report of the manuscript corrections. All changes are visible through track changes modality on word doc attached.
- With respect to the advice of separating the topic of AVMs and fistulas, we choose to include these vascular lesions in one folder precisely for the purpose of a comprehensive view of the use of such a versatile toll as the microvascular doppler sonography. As you suggested, we stressed the concept of combining different tools to enhance the surgical results in vascular neurosurgery, in our conclusions.
- Finally, we attempted to check all ortographic mistakes in the text.
Many thanks in advance
Best regards

Reviewer 2 Report
Comments and Suggestions for Authors
Dear Editor,
I read with great care the referenced manuscript titled “The Use of Intraoperative Microvascular Doppler in Vascular Neurosurgery: Rationale and Results. A Systematic Review” which has been submitted for publication in BSJ.
The aim of the study was to review the role of intraoperative micro vascular Doppler ultrasound in vascular neurosurgery. The search question is original and interesting. The literature is well conducted.
However, the study has some important limitations.
- All abbreviations should be explained when initially used, and thereon should be consistently used.
- The study needs to define the outcomes of interest, which are the endpoints of the review.
- The quality of the eligible studies should be assessed.
- Table 1 is a useful. However, it should be made easy-to-read.
- A limitations section should be added.
From all the above, it becomes obvious that the current manuscript draft can be improved with a minor revision.
Best regards
Comments on the Quality of English LanguageThe music rip t is well written. A few typos should e corrected.
Reviewer 3 Report
Comments and Suggestions for Authors
This is an interesting systematic review, with valuable insights on the use of several tools for intraoperative vessel flow assess during vascular microneurosurgery. Even if the novelty of the information is not striking, this review can be useful as an overview of this topic.
I suggest the following point of revision:
Discussion:The paragraph about aneurysm surgery seems too long. I suggest to shorten it. For instance, the part concerning the comparison of time consumption between Microdoppler and Digital Angiography is redundant.
Conclusions: After fairly discussing the efficacy and limitations of Microdoppler in aneurism surgery, the conclusions seem unbalanced in favor of this techniques. I suggest to underscore, in this section also, the need of a multimodal approach to vessel patency checking, in which different techniques can be complimentary, rather than exclusive. From this review, a combined use of Microdoppler and ICG-Video-angiography appears to be a reasonable practice for most microsurgical vascular procedures.
Minor reviews:
Line 77-78: It is not clear why V.G. is an "independent" Author.
Line 244-245: please, check this sentence; a verb is probably missing.
Line 337: please, check "resident".
Comments on the Quality of English LanguageOverall English language is good. I suggest some style revision by an English language expert.
Reviewer 4 Report
Comments and Suggestions for Authors
The authors reviewed the clinical utility of intraoperative microdoppler in surgeries for aneurysms, AVMs, and dural AVFs. Its usefulness has now become obvious in vascular neurosurgery and the content of this review is not so new to the readers. However, it is important to sort out articles available.
There are no points to add in my opinion, but there are some typographical errors. So please correct them.
Comments on the Quality of English Language
There are many typographical errors. The manuscript should be edited.
